

# The significance of biowaste drying analysis as a key pre-treatment for transforming it into a sustainable biomass feedstock

Fernando Damián Barajas Godoy[1], Marco A. Martínez-Cinco[1], José G. Rutiaga-Quiñones[2], Otoniel Buenrostro-Delgado[3] and Jose Mendoza[1]

[1] Facultad de Ingeniería Química, Universidad Michoacana de San Nicolás de Hidalgo, Morelia, Michoacan, Mexico
[2] Facultad de Ingeniería en Tecnología de la Madera, Universidad Michoacana de San Nicolás de Hidalgo, Morelia, Michoacan, Mexico
[3] Instituto de Investigaciones en Ciencias de la Tierra, Universidad Michoacana de San Nicolás de Hidalgo, Morelia, Michoacan, Mexico

Corresponding authors
Marco A. Martínez-Cinco,
marco.martinez@umich.mx
Otoniel Buenrostro-Delgado,
otoniel.buenrostro@umich.mx

## ABSTRACT

The objective of this study is to investigate the drying kinetics of fruit and vegetable peel biowaste using a sustainable technique as a key-pretreatment for its conversion into useful feedstock. Biowaste represents a missed potential source of bioenergy and bioproducts, but moisture removal is required, and conventional drying methods are expensive since they require great quantity of energy supplied, almost always, by a non-renewable energy. In this study six batches with the same quantity of biowaste, and heterogeneous physical composition were dried under open-sun conditions. We evaluated the influence of the interaction between drying area and the initial moisture content on drying rate. Eight semi-theoretical models were fitted using Levenberg–Marquardt algorithm to predict drying rate, and their accuracy was assessed through goodness-of-fit tests. Maximum moisture content to preserve biomass (10%) was reached on 5th day and the equilibrium on 16th day of drying. According to goodness-of-fit test ($R^2 = 0.999$, $\chi^2 = 4.666 \times 10^{-5}$, RMSE $= 0.00683$) the best model to predict drying rate was Two-term model. The mathematical model obtained from Fick's second law is reliable to predict drying kinetics, $R^2$ ($0.9648 \pm 0.0106$); despite the variation between drying area and initial moisture content. Kruskal-Wallis test showed that drying rates between batches are not significantly different ($p = 0.639$; 0.05); nor effective diffusion coefficient ($D_{eff} = 4.97 \times 10^{-11} \pm 0.3491 \times 10^{-11}$), ($p = 0.723$; 0.05). The study of drying kinetics is crucial for selecting the optimal biowaste treatment based on its generation context. This could enable its use as feedstock for bioproduct or bioenergy production, thereby reducing waste accumulation in landfills and environmental impact.

## INTRODUCTION

Municipal solid waste (MSW) is a global concern topic (*Ayeleru, Okonta & Ntuli, 2018*; *Tumuluru, Yancey & Kane, 2021*; *Jakubus & Spychalski, 2022*; *Czatzkowska et al., 2023*; *Tumu, Vorst & Curtzwiler, 2023*), it is linked to linear economy ('take-make-waste') paradigm (*MacArthur, 2024*). "In 2020, the world was estimated to generate 2.24 billion tons of solid waste", it is expected to increase from 2020 levels to 3.8 billion tons in 2025 (*World Bank Group, 2024*). The world production of MSW is 1300 million tons per year, and projections for 2025 of the increase in MSW generation will reach 2200 million tons (*Lucian et al., 2018*). MSW are mainly composed by biowaste (34%), and that approximately 86 million tons comes from uneaten food (*EEA, 2020*). In the European Union (EU), landfilling of biowaste is the least desirable option, however around 40% is landfilled, and in most of developing countries, still 100% is landfilled. In the United States (US), nearly 50% of MSW is landfilled, food waste accounts for approximately 20% of MSW, and MSW landfills are the third-largest source of methane emissions (*EPA, 2023*). An increase in MSW generation in Latin America has been associated with population growth. In 2015, the generation of MSW in Mexico was 53.1 million tons; which represents an increase of 73% between 2000 and 2015 (*Salazar, 2021*). In Morelia, capital of the state of Michoacan, 700 tons of MSW are produced daily, and 81% is organic matter generated from different sources and varied decomposition rates (*Hernández et al., 2019*). Dumping MSW in landfills remains as the only alternative, due to a lack of founds to properly build and manage landfills, the above impede installation of geomembranes an essential component that prevents leachates from seeping into ground, and allows methane and carbon dioxide to escape, which are the foremost causative greenhouse effect (*Osman et al., 2019*; *EPA, 2023*). On the other hand, closing dumpsites is a top priority for the International Solid Waste Association (ISWA). This is not only because of environmental impacts but also due to damage to health and violation of human rights associated with these issues (*ISWA, 2024*). The issue is landfills do not comply with the requirements to apply and prevent the escape of leachate and biogas mainly (*Ortner, Müller & Bockreis, 2013*). Biowaste has been used as a composting material, however, not all waste byproducts are suitable because some of them can affect biodegradation process by inhibiting decomposing bacteria growth and also because some variables, such as the temperature reached during the process, are insufficient to eliminate some microorganisms that are pathogenic to humans (*Mavaddati et al., 2010*). Compost can also cause negative impacts in soil, by increasing toxic substances and nitrates presence, which depending the concentration can be toxic to plants and animals, and even cause damage in food chain (*Fagnano et al., 2011*). Instead, the organic fraction of MSW accumulates in dumps. Nowadays, it is mandatory to use it as a source of bioenergy (*Unpinit et al., 2015*; *Pavi et al., 2017*). Diverse studies have been proposed alternative solutions to new challenges that arise in the management of MSW, one of the most feasible proposals is the conversion to bio-digesters, since they effectively stabilize the organic fraction. A thorough comprehension of MSW generation can offer valuable insights to authorities regarding optimal management practices (*Kathiravale et al., 2003*; *Ayeleru, Okonta & Ntuli, 2018*). A significant amount of biowaste within MSW

emphasizes the need for developing technologies that enable efficient utilization of natural resources. These technologies must promote environmental and socio-economic benefits that accelerate the transition from traditional waste management to waste recycling. Choosing the optimal method and technology for managing MSW depends on specific conditions and needs of site where it is generated. This decision should be backed by an assessment of environmental impacts and energy benefits through a life cycle analysis (LCA). This analysis compares various scenarios based on available technologies, aiming to prevent MSW disposal and recover energy from it (*Jeswani & Azapagic, 2016*). To prevent the negative impacts of MSW dumping, it is imperative to valorize them as value-added byproducts to obtain bioenergy and bioproducts (*EEA, 2020*). The high moisture content of biowaste (75% to 95%) increases the cost of MSW management, with up to 60% of the budget spent on collection and transportation. Therefore, drying emerges as a viable alternative to reduce management costs, regardless of whether the byproducts are valorized (*Konstantzos et al., 2019*). Waste-to energy conversion technologies have low energy efficiency due to variable moisture content in MSW; 5–15% moisture content is desirable for co-firing, gasification, and pyrolysis (*Tumuluru, Yancey & Kane, 2021*). Typically, biomass waste with less than 30% moisture content is considered a dry feedstock and above 30% is considered wet feedstock (*Wang & Tester, 2023*). A crucial factor in valuing certain types of biowaste is their high moisture content. For example, melon rinds can contain up to 90% water, making it essential to dry them. When biomass is used for producing biofuels like pellets and briquettes, its moisture content should be 8–10% for pellets and 15–30% for briquettes. Higher moisture levels compromise the integrity and stability of the biofuel and reduce its combustion temperature. Moisture content also affects the production of bioproducts (*Kamdem et al., 2015*; *Dinesha, Kumar & Rosen, 2019*). Even, high moisture content affect negatively in total fuel calorific power and combustion efficiency (*Pradhan, Mahajani & Arora, 2018*). Since biomass naturally contains high moisture, this factor becomes the main obstacle to valorize MSW with high heating potential (*Cai et al., 2017*). Biomass high moisture content requires elevated energy consumption, and longer drying periods (*Dinesha, Kumar & Rosen, 2019*; *Konstantzos et al., 2019*). Drying costs and kinetics depend on method and drying conditions applied (*Gaibor et al., 2016*). Drying materials is a complex process that involves energy and mass transfer simultaneously, which occur inside the wet material (*Sobukola et al., 2007*). During heat transfer, moisture is removed as vapor from the material's surface and depends on external conditions such as temperature, humidity, air velocity, exposed surface area, and pressure. Meanwhile, movement of matter is due to moisture migration within the solid and depends on its physical nature, diffusion rate, and moisture content (*Tiwari, 2016*). If heat transfer occurs at a significantly higher rate than mass transfer during drying of material, drying rate is primarily determined by how quickly water moves from within to the surface of solid, therefore moisture diffusion is use to describe mass transfer process (*Ozdemir & Devres, 1999*). Fick's second law is one of the most widely used mathematical models to predict the drying kinetics of food and agricultural products, both for solids with spherical and slab geometry. In which the predominant mechanism in the movement of water within the solid occurs by diffusion (*El-Amin, 2011*). When maintaining constant

conditions during drying is possible, three characteristic periods of drying rate typically occur: (a) adjusting rate, (b) constant rate, and (c) falling rate (*Treybal, 1981*). "Thin layer drying (TLD) means to dry as one layer of sample particle or slice" (*Sobukola et al., 2007*). The application of semi-theoretical mathematical models in TLD have demonstrated their effectiveness in predicting the drying kinetics in the descending falling rate period (*Badaoui et al., 2019*). When drying occurs under weather conditions, the process may be affected by variability of this conditions throughout the day, because of this phenomenon, the drying rate under these circumstances is not constant; nevertheless, it is considered the cheapest way to reduce MSW moisture (*Chen, Chang & Lee, 2015*). Solar energy can play an active role in meeting the energy demand, as a friendly environment way to reduce the drying process cost, as mentioned by *Badaoui et al. (2019)*. Data generation for drying kinetics analysis of waste is important in order to use specific technology to dry them and establish conditions for an efficient operation. This allows the dried product to reach a desired moisture content over time, serving as a pretreatment of waste and enabling their use as feedstock in biorefining, where a specific moisture content is required to convert them into bioenergy or bioproducts (*Reyhanitabar et al., 2020*; *Eixenberger et al., 2024*). The objective of this study is to investigate the drying kinetics of fruit and vegetable peel biowaste (FVBW) by sun-drying, as a sustainable key-pretreatment for its conversion into useful feedstock.

## MATERIALS & METHODS

### Biowaste sampling

Sampling was carried out in Morelia, Michoacan, Mexico. Despite the fact that Morelia city has regulations for solid waste management, it is not complied by any of the actors involved, which also makes it difficult to collect samples (*Hernández et al., 2019*). The sampling area was chosen randomly without considering the economic stratum, as defined in Mexico by the National Institute of Statistics, Geography and Information (INEGI), which categorizes income into four ranges: Low, Lower-Middle, Upper-Middle, and High Stratum (*INEGI, 2017*). The selected area falls into lower-middle stratum. Samples were collected from municipal solid waste collection trucks with an average weight of five kilograms (composed of fresh fruit and vegetable peels), ensuring that samples were not tampered with and that valuable materials were extracted. Biowaste sampling was conducted on six consecutive days during the third week of May 2019, and was refrigerated at 4 °C. Samples were divided into six batches based on collection day, with each batch containing three kilograms of FVBW. To homogenize sample particle sizes, waste greater than 0.0254 m width, 0.0508 m length, and 0.005 m thickness, was sliced according to *NREL/TP-510-42621 (2008)*.

### Drying of biowaste samples

Six batch of FVBW formed during sampling stage, was spread evenly on a metal surface measuring 0.9 m long by 0.5 m wide and dried under open-sun conditions for 10 h (from 9:00 a.m. to 7:00 p.m.). Drying process was performed during 16 days between June and July (2019). In that period weather conditions were: maximum relative humidity of

**Table 1** Semi-theoretical mathematical drying models tested.

| Model No. | Model name | Equation | References |
|---|---|---|---|
| 1 | Handerson&Pabis | $MR = a \cdot e^{-k \cdot t}$ | *Henderson & Pabis (1961)* |
| 2 | Newton/Lewis | $MR = e^{-k \cdot t}$ | *Lewis (1921)* |
| 3 | Page | $MR = e^{-k \cdot t^n}$ | *Page (1949)* |
| 4 | Parabolic Model | $MR = a + b \cdot t + c \cdot t^2$ | *Sharma & Prasad (2004)* |
| 5 | Logarithmic | $MR = a \cdot e^{-k \cdot t} + c$ | *Yaldyz & Ertekyn (2001)* |
| 6 | Two-Term | $MR = a \cdot e^{-k_0 \cdot t} + b \cdot e^{-k_1 \cdot t}$ | *Henderson (1974)* |
| 7 | Wang & Singh | $MR = 1 + a \cdot t + b \cdot t^2$ | *Ozdemir & Devres (1999)* |
| 8 | Midilli | $MR = a \cdot e^{-k \cdot t^n} + b \cdot t$ | *Midilli, Kucuk & Yapar (2002)* |

Notes.
a, b, c, n dimensionless drying constants; k, $k_o$, $k_1$, drying rate constants (1/h)

77.25%, temperature of 47 °C (direct sunlight exposition) and maximum wind velocity of 19.3 km/h. Each batch was weighed daily on a 5 kg digital scale with 0.01 g accuracy. For all batches, under these conditions each drying curve only exhibits the second falling rate period at environmental conditions, and the drying process occurs in a non-steady state. Given the nature of biomass that makes up each batch of FVBW, the initial moisture content ($M_o$) differs between batches as well as the drying exposed area (**A**). Dry mass for each batch was determined according to *ASTM D 4442-92 (1992)*.

## Influence of initial moisture content and drying exposed area on drying rate

The influence of **A** and $M_o$ interaction on drying rate (**N**) was analyzed using a surface model for each batch (Eq. 1):

$$N(A, M_0) = a_1 A^2 + a_2 A \cdot M_0 + a_3 M_0^2 + a_4 A + a_5 M_0 + a_6. \tag{1}$$

For each batch, each constant of each model was calculated in three experimental key moments, using GNU Octave version 7.1.0[®] software (*Eaton et al., 2022*): (1) initial conditions, when FVBW presented maximum moisture content; (2) on the fifth day, when FVBW reached the preservation optimum moisture content (10%); and (3) at the end of the experiment, when the equilibrium moisture content was reached. The coefficient of determination ($R^2$) was calculated for each model to test the effectiveness of **N** (**A**, $M_o$) in predicting experimental data.

## Biowaste mathematical drying models

The effectiveness of eight models in predicting drying rates was assessed, which have previously been applied to other types of biomass: (*Arslan & Musa, 2010*) in onion slice drying, (*Erbay & Filiz, 2010*) thin layer food drying studies, (*Gaibor et al., 2016*) maize cane drying, (*Inyang, Oboh & Etuk, 2018*) modeling food drying techniques, and (*Wakchaure et al., 2010*) mushroom (*Agaricus bisporus*) drying. Mathematical models presented in Table 1 were fitted from experimental results of moisture ratio *vs.* time ($t$); dimensionless model constants $a,b,c,d$ and n and drying constants $k$, $k_o$ and $k_1$ ($h^{-1}$) were estimated by nonlinear least squares using the Levenberg–Marquardt algorithm (*Levenberg, 1944*; *Gavin, 2022*).

## Effective diffusivity coefficient

Fick's second law in one dimension was applied (*Crank, 1975*) (Eq. (2))

$$\frac{\partial MR}{\partial t} = D_{eff} \frac{\partial^2 MR}{\partial x^2} \tag{2}$$

where $MR$ is dimensionless moisture ratio; $t$ drying time; $x$ distance in the moisture transport direction; and $D_{eff}$, effective diffusivity coefficient.

For a rectangular slab with thickness of 2L and symmetrical plane at $x = 0$; the initial (*i*) and boundary (*ii*, *iii*) conditions are:

$(i)\, t = 0, 0 < x < L, M = M_0$
$(ii)\, t > 0, x = 0, \frac{\partial M}{\partial x} = 0$
$(iii)\, t > 0, x = L, M = M_e$

where $M_0$: indicates the initial moisture content; $M_e$: equilibrium moisture content and $M$: moisture content at any time.

Equation (2) is solved by power series; when internal mass transfer is the controlling transport mechanism in one dimension, assuming an infinite slab (Eq. (3)) (*Crank, 1975*; *Derossi, Severini & Cassi, 2011*; *Cai et al., 2017*).

$$MR = \frac{8}{\pi^2} \sum_{i=0}^{\infty} \frac{1}{(2i+1)^2} \exp[\frac{-\pi^2(2i+1)^2}{4L_0^2} D_{eff} \cdot t]. \tag{3}$$

For long drying periods, when $i = 0$ and the drying process is carried out under unsteady-state conditions, Eq. (3) can be further simplified to only the first term of the series (Eq. (4)) (*Pisalkar et al., 2014*).

$$MR = \frac{8}{\pi^2} \exp[\frac{-\pi^2}{4L_0^2} D_{eff} \cdot t]. \tag{4}$$

According to *Lewis (1921)*; $MR$ is expressed as Eq. (5):

$$MR = \frac{(M - M_e)}{(M_0 - M_e)}. \tag{5}$$

Substituting $MR$ from the Eq. (5) in Eq. (4), Eq. (6) is obtained:

$$MR = \frac{(M - M_e)}{(M_0 - M_e)} = \frac{8}{\pi^2} \exp[\frac{-\pi^2}{4L_0^2} D_{eff} \cdot t]. \tag{6}$$

Finally, when logarithms are applied on Eq. (6), Eq. (7) is obtained:

$$lnMR = ln[\frac{8}{\pi^2}] - [\frac{\pi^2}{4L_0^2} D_{eff} \cdot t]. \tag{7}$$

## Statistical analysis

The Fickian model and semi-theoretical mathematical models presented in Table 1 were assessed using goodness-of-fit tests; coefficient of determination ($R^2$), reduced Chi-square ($\chi^2$), and root mean square error (RMSE) were calculated for each model. Highest $R^2$

**Table 2  Drying area and initial moisture content (d.b.) in every batch.**

| | Batches | | | | | |
|---|---|---|---|---|---|---|
| Variable | B1 | B2 | B3 | B4 | B5 | B6 |
| $A$ (m$^2$ ) | 0.8336 | 0.9859 | 0.7912 | 0.8300 | 0.8400 | 0.7601 |
| $M_o$(Kg$_{H2O}$/Kg$_{SS}$) | 3.39 | 3.18 | 3.02 | 2.92 | 2.99 | 2.64 |

Notes.
$Kg_{H2O}$, water kilograms; $Kg_{SS}$, dry solid kilograms; Bi, batch i ($i = 1:6$); d.b., dry basis Kg moisture/Kg dry solid.

**Table 3  Accuracy of $N(A, M_o)$ and its constants, at three key moments of drying.**

| Day | $a_1$ | $a_2$ | $a_3$ | $a_4$ | $a_5$ | $a_6$ | $R^2$ |
|---|---|---|---|---|---|---|---|
| 1 | 0.0739 | 1.0227 | −1.1974 | −0.3850 | −1.1357 | 1.0924 | 0.999 |
| 5 | −0.0160 | 0.1773 | −0.1822 | −0.0478 | −0.2477 | 0.1875 | 0.999 |
| 16 | −0.0021 | 0.0318 | −0.0340 | −0.0134 | −0.0423 | 0.0408 | 0.999 |

Notes.
$a_1$, $a_2$, $a_3$, $a_4$, $a_5$, and $a_6$ are surface model constants.

values and lowest $\chi^2$, and RMSE values were used as criteria to define the effectiveness of the best model in predicting experimental data.

The Kruskal-Wallis test ($df = 5$ and $\alpha = 0.05$) was used to compare the mean values of **N** among the six batches with respect to **A** and $M_o$.

## RESULTS

### Characterization of biowaste samples based on drying area and initial moisture

The components of FVBW within and among batches, were heterogeneous respect to **A** and $M_o$ (Table 2).

Drying process at constant rate, **A** and $M_o$ are inversely related, this suggests that batches with higher values of $M_o$ may take longer to dry; according to $M_o$ values of batch one (B1) and batch two (B2) samples, it can be inferred that it takes a longer drying time than the rest of batches; however, in this work, where drying process took place under unsteady-state, the drying rate exhibits a different response to $M_o$.

### Behavior of drying rate and its relationship with the initial moisture and drying area

Every surface mathematical models $N(A, M_o)$, presented R$^2$ values of 0.999. Which demonstrated the effectiveness of $N(A, M_o)$ to predict **N** values with high accuracy at three key moments of the experiment (Table 3).

At initial conditions of drying process, interaction among the experimental data of **A** with $M_o$ and its influence on **N**, B6 sample reached the highest value of **N**, which is associated with the lowest values of **A** and $M_o$ (Fig. 1A). The behavior of **N**, estimated as a function of **A** and $M_o$, from the surface mathematical model, is increasing in the maximum

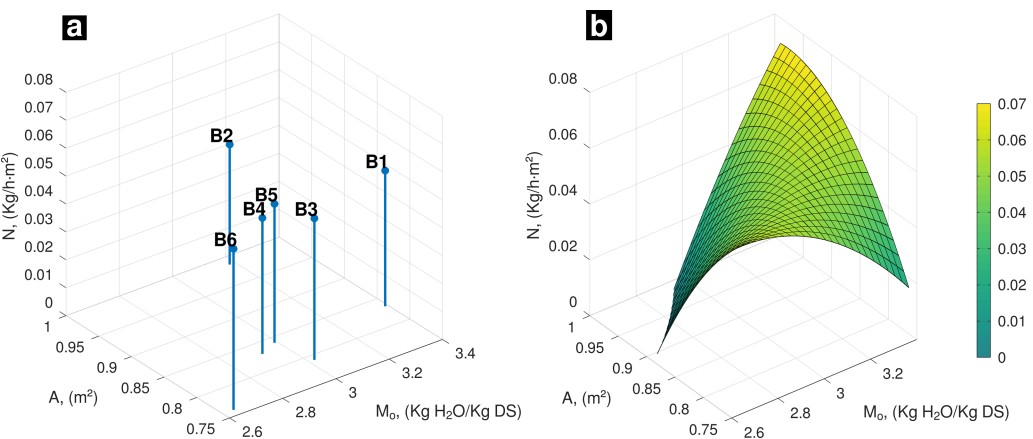

**Figure 1** (A) Experimental data, influence of A and $M_o$ on drying rate; (B) Behavior of $N$ $(A, M_o)$ on day one.

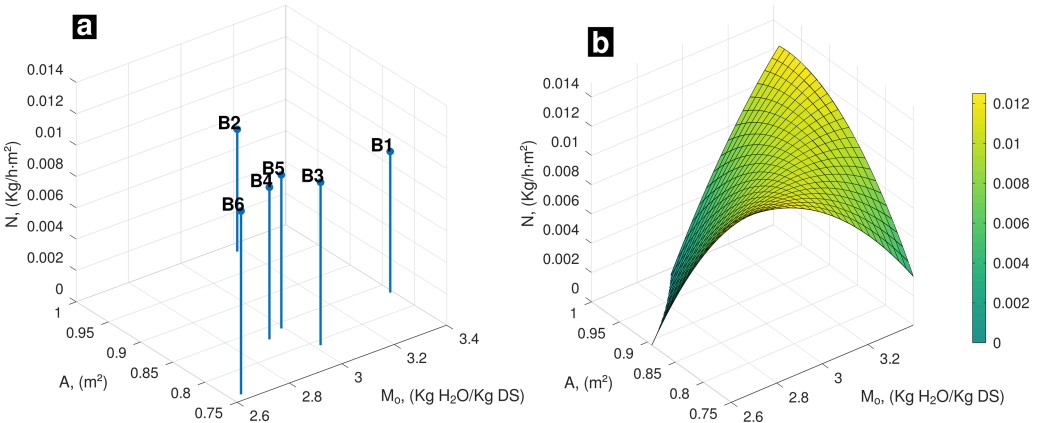

**Figure 2** (A) Experimental data, influence of $A$ and $M_o$ on drying rate; (B) Behavior of $N$ $(A, M_o)$ on the fifth day of drying.

values of **A** and $M_o$ as well as at the opposite side associated with minimum values of **A** and $M_o$ (Fig. 1B).

In Fig. 2A, results of **N** with respect to **A** and $M_0$ at fifth day are presented. B6 kept highest **N** value. At this moment, **N** trending observed is similar that at experiment's start (Fig. 2B). Biggest **N** values appear at highest interactions between **A** and $M_0$, as well as opposite side where batches presented lowest **A** and $M_0$ values.

At final stage of experiment (day 16), FVBW samples reached equilibrium moisture (Fig. 3A). **N** follows same pattern as observed at the first and fifth day (Fig. 3B). In general, it is observed that highest values of **A** associated with highest values of $M_o$ promote **N** to

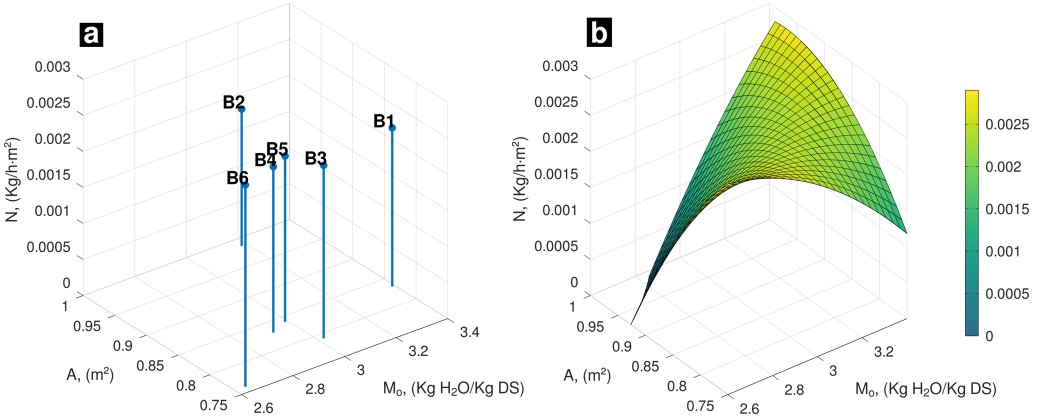

**Figure 3** (A) Experimental data, influence of $A$ and $M_o$ on drying rate; (B) behavior of $N$ $(A, M_o)$ on the 16th day of drying.

**Table 4** Drying rate and MR at first, fifth and 16th drying day.

| Day | N, (Kg/m² h) (± Std. Dev.) | Moisture removed (%) |
|---|---|---|
| 1 | 0.0498 ± 0.0047 | 60.1 |
| 5 | 0.0075 ± 0.0010 | 94.3 |
| 16 | 0.0023 ± 0.0003 | 99.9 |

reach highest values. Analogously, same behavior tends to occur at opposite conditions where lowest values of **A** are associated with lowest values of $M_o$.

The range within which the **N** values appear among batches and their means, for each day, shows low dispersion, according to obtained values of the standard deviation (Table 4). After five days of treatment the MR value is greater than 90%, half of the time required to dry cylindrical bodies of corn cane previously reported by *Gaibor et al. (2016)*. This moisture content is within the desired range to prevent the development of biomass-decomposing bacteria; moreover, within this moisture range, the biomass can enter bio-refinery processes such as pyrolysis or densification of biomass (*Bajwa et al., 2018*).

## Drying curves and semi-theoretical drying models

In Fig. 4A, results of **N** *vs. t* are presented, batch six (B6) sample presented the higher values of **N** followed by batch three (B3) sample. It is important to mention that initial moisture content and dry solid mass were variable among batches. On the fifth day, B3 sample presented the highest value of remaining moisture content; and batch five (B5) sample the lowest (Fig. 4B); in spite of this findings, when Kruskal-Wallis test was applied, significant differences were not found ($p = 0.639$, $df = 5$ and $\alpha = 0.05$).

The experimental MR results among batches differed by 0.06% when the initial moisture content in samples was present; 0.02% when moisture content reach 10% value; and

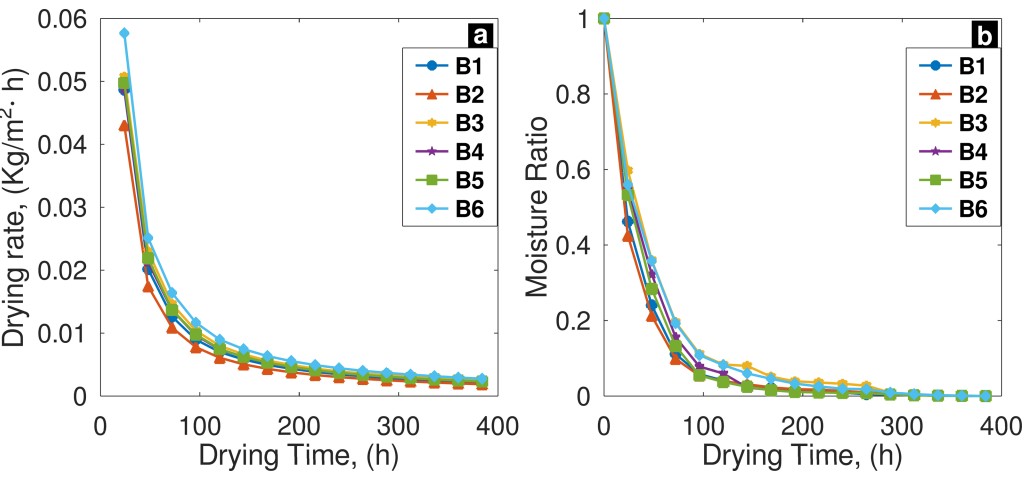

**Figure 4** (A) Drying rate trend; and (B) moisture ratio behavior among batches, only falling rate period occur.

**Table 5** Evaluated semi-theoretical models and their constants. Bold numbers indicate the best model (two-term) for predicting the drying rate.

| Model | Constants | $R^2$ | $\chi^2$ | RMSE |
|---|---|---|---|---|
| 1 | $a = 1.8474$<br>$k = 0.6195$ | 0.998 | $1.2281 \times 10^{-04}$ | 0.0110 |
| 2 | $k = 0.3668$ | 0.894 | $7.2452 \times 10^{-03}$ | 0.0851 |
| 3 | $a = 0.8393$<br>$b = -0.1573$<br>$c = 0.0067$ | 0.799 | $1.5706 \times 10^{-02}$ | 0.1253 |
| 4 | $n = 2.0013$<br>$k = 0.1301$ | 0.974 | $1.8956 \times 10^{-03}$ | 0.0435 |
| 5 | $k = 0.6393$<br>$a = 1.8723$<br>$c = 0.0092$ | 0.999 | $6.8903 \times 10^{-05}$ | 0.0083 |
| 6 | $a = 1.8664$<br>$b = 0.0669$<br>$k_o = 0.6823$<br>$k_1 = 0.1678$ | **0.0999** | **$4.06660 \times 10^{-05}$** | **0.00068** |
| 7 | $a = -0.1940$<br>$b = 0.0085$ | 0.764 | $1.7204 \times 10^{-02}$ | 0.1312 |
| 8 | $a = 2.2117$<br>$b = 3.891 \times 10^{-4}$<br>$k = 0.7936$<br>$n = 0.8664$ | 0.999 | $7.1463 \times 10^{-05}$ | 0.0084 |

0.00032% when drying process reach the equilibrium. According to goodness of fit test (Table 5), Two-term model presented best fitting to predict drying rate ($R^2 = 0.999$, $\chi^2 = 4.666 \times 10^{-5}$ and RMSE $= 0.00683$).

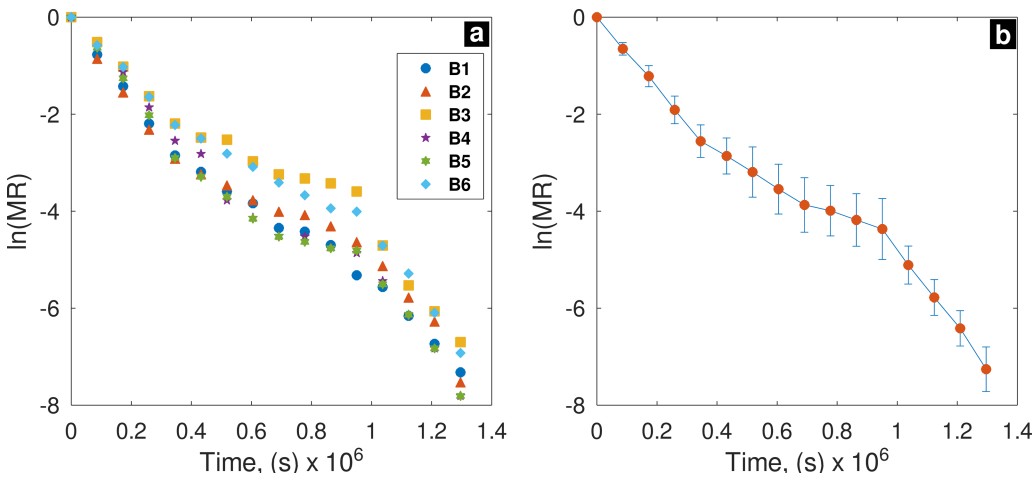

**Figure 5** *ln (MR) vs.* drying time (*t*); (B) average and error bar of *ln (MR).*

**Table 6 Coefficient of determination and effective diffusivity coefficient.**

| Parameter | Batches | | | | | |
| --- | --- | --- | --- | --- | --- | --- |
| | B1 | B2 | B3 | B4 | B5 | B6 |
| $R^2$ | 0.979 | 0.952 | 0.954 | 0.970 | 0.962 | 0.972 |
| Slope ($\times 10^{-06}$) | −5.05 | −4.64 | −4.54 | −5.35 | −5.23 | −4.65 |
| $D_{eff}$ ($\times 10^{-11}$) | 5.11 | 4.70 | 4.59 | 5.42 | 5.29 | 4.71 |

## Determination of effective diffusivity coefficient

In Fig. 5A, results of *ln (MR) vs. t* are shown. Different points in the graphic represent six different batches; $D_{eff}$ is obtained for the curve slope of *ln (MR)* and *t* relationship (Fig. 5B). Kruskal-Wallis test showed that there is no significant difference ($p = 0.723$, $\alpha = 0.05$) of $D_{eff}$ among batches.

In all batches $R^2$ values were above of 95%. B1 sample presented the highest $R^2$ value (0.979), and B2 sample the lowest (0.952); despite of drying process occurred in unsteady-state, this finding demonstrate a significant linear relationship between *ln(MR)* and *t* (Table 6); hence, the model obtained from Fick's second law is reliable to predict **N** values for biowaste samples under open sun-drying conditions.

## DISCUSSION

### Initial moisture content and heterogeneous physical composition of samples

The initial moisture content in FVBW is relatively higher than that of forest and agricultural biowaste (*Libra et al., 2011*; *Pavi et al., 2017*; *Osman et al., 2019*; *Qu et al., 2021*; *Wang & Tester, 2023*; *Eixenberger et al., 2024*). This implies a greater difference between $M_0$ and $M_e$, which, along with meteorological conditions, is a critical factor in the decrease of the drying rate. The drying rate slowed as the moisture content approached equilibrium moisture, as reported by *Zhen-dong et al. (2022)* in the direct sun drying of corn seeds.

This suggests that the drying period to reach the same final moisture content in FVBW would be longer in samples where the difference between $M_0$ and $M_e$ is greater, implying additional energy consumption, as mentioned by *Kaveh, Abbaspour-Gilandeh & Fatemi (2021)*. Heterogeneous nature of physical composition of FVBW affected $M_0$ in each batch, resulting in $M_0$ behaving randomly during sampling. However, $M_0$ showed low variation among batch values, allowing for sufficient control over $M_0$ to achieve uniform drying operation. When considering the projection of solar drying on a pilot scale, $M_0$ values would be similar, and an effective treatment could be achieved. Based on this, final moisture content within a specific drying period would be consistent, in line with the observed values of *MR* and its standard deviation at initial conditions, after five days, and after 16 days of treatment.

### Effectiveness of surface model and correlation of $M_0$ and A with maximum N

Drying rate and dimensionless moisture content are typically reported as functions of time; however, $N$ has a multivariable dependence (*Treybal, 1981*). In this study, it was demonstrated that the surface model $N(M_0, \mathbf{A})$ is effective and highly accurate predicting $N$ as a function of $M_0$ and $\mathbf{A}$. Through this model, it was identified that maximum drying rate of FVBW is correlated with higher values of $M_0$ and $\mathbf{A}$. It is of particular interest to observe the response of $N$ as a function of $M_0$ and $\mathbf{A}$ since, on one hand, $M_0$ varies between batches and is expected to do so each time FVBW is collected due to its origin; on the other hand, the drying area is a variable that can be relatively easier to control to achieve more uniform treatment and greater control over the operation. The response of $N$ at the beginning of the experiment, when the value of $M = 10\%$ is reached, and when equilibrium is reached, demonstrated that the behavior of $N(M_0, \mathbf{A})$ remains consistent over time. This supports the assertion that drying method is effective despite being conducted under non-constant drying conditions and that the mechanisms involved in the movement of water from the inner part of the solid to the surface are similar among samples from the six batches.

### Drying curves and the controlling mechanism of drying kinetics

When the controlling mechanism of drying kinetics is due to internal factors within the solid, the properties of the drying air and its velocity become insignificant (*El-Amin, 2011*). If the controlling mechanism of drying kinetics is due to internal factors, the internal morphology of the solid influences the trend of the drying curve that characterizes the drying of the solid (*Zhao et al., 2023*). The close similarity between the batches in the trends of drying curves $N$ *vs* $t$ (Fig. 4A), and *MR* *vs* $t$ (Fig. 4B), suggests that there is sufficient resemblance in the arrangement of the channels and voids that define internal structure of substrate. This means that the path that water molecules have to take in order to move from the inner part of the substrate to the surface would generally encounter the same difficulty across the six batches of FVBW. This fact is reflected in the similarity among the drying curves of the six batches and demonstrates the repeatability and efficacy of the method. The trend of drying-curves is similar to those reported by *Agbede et al. (2023)* in the drying

of banana stalks, dried under sun drying. As seen in the results of the *MR vs t* curves, discrepancies in *MR* among the batches over time are small. In these curves, it is more notable that these discrepancies are related to differences between $M_o$ and $M_e$. This fact demonstrated that, despite these discrepancies, drying remained uniform over time and become more evident after *MR* reached 100 h of drying and approached equilibrium. This trend is consistent with other studies in which agricultural products and by-products were dried under sun-drying (*Suherman et al., 2020*; *Simo-Tagne & Ndukwu, 2021*; *Zhen-dong et al., 2022*; *Zhao et al., 2023*). Sun-drying is the most economical, environmentally friendly, and widely used technique for drying agricultural products and by-products (*Zhao et al., 2023*). However, to handle the daily volume of FVBW generated, it is necessary to incorporate an environmentally friendly technological adaptation to the method to reduce the moisture content just below 10% within a few hours, instead of five days, as other researchers have successfully done in drying other types of biomass using solar energy (*El-Sebaii & Shalaby, 2012*; *Badaoui et al., 2019*; *Devan et al., 2020*).

## Drying models and their response under non-constant drying conditions

The drying models evaluated in this study have been successfully employed in sun-drying operations, as well as under constant drying conditions, and in drying thin layers of agricultural and food products (*Henderson, 1974*; *Erenturk, Gulaboglu & Gultekin, 2004*; *Sobukola et al., 2007*; *Gaibor et al., 2016*; *Goud et al., 2021*; *Dhake et al., 2023*). These models have proven to be particularly useful predicting drying rate in falling rate period (*Henderson, 1974*). Consistent with the above, in present study, Logarithmic, Two-Term, and Midilli models provided best fit to experimental data, with as much effectiveness and precision as reported in the drying operations of food and agricultural products. *Sobukola et al. (2007)* dried leafy vegetables under direct sunlight and found that the Logarithmic and Midilli models were the most reliable, with the Midilli model being the most accurate ($R^2 = 0.999$). *Badaoui et al. (2019)* dried tomato residues in a greenhouse-type solar dryer and, among conventional models they evaluated, Midilli-Kucuk model achieved an $R^2$ value of 0.999, while Logarithmic model achieved an $R^2$ of 0.998. *Castillo-Téllez et al. (2024)* dried oregano leaves directly under sun and found that Modified Page, Page, and Logarithmic models provided best fit, with Logarithmic model being the most reliable ($R^2 = 0.998$). *Suherman et al. (2020)* dried cassava flour in a hybrid solar dryer and found that Midilli model was the most accurate ($R^2 = 0.994$). Logarithmic and Two-Term models proved to be the best for predicting drying rate of rosehip under constant conditions, with Logarithmic model achieving an $R^2$ value of 0.994 (*Erenturk, Gulaboglu & Gultekin, 2004*). The *MR vs t* curve exhibits a trend similar to that indicated by *Henderson (1974)*, suggesting that the mechanism controlling water transport is diffusion. Given that the effective diffusivity coefficient is sensitive to temperature (*Baldán et al., 2020*), it is inferred that temperature changes throughout the day affected dispersion of *ln(MR) vs. t* relationship results, particularly when *MR* approached 10%. However, the $D_{eff}$ values showed low dispersion according to their standard deviation. Moreover, values obtained from six batches fall

within the acceptable range of $D_{eff}$ values for agricultural products and food, which are between $10^{-11}$ and $10^{-9}$ m$^2$/s (*Sobukola et al., 2007*).

## CONCLUSION

A small statistical discrepancy in $M$ values between batches of FVBW demonstrated effectiveness of this drying method despite variation in physical composition among batches and varying drying conditions throughout the day and over the days of the experiment. No significant differences were found in $N$, $D_{eff}$, and $MR$ based on interaction of $A$ and $M_0$ among batches. Although drying method is effective and environmentally friendly, it is imperative to scale and adapt it to reduce drying time and address the daily generation of FVBW. The data generated in this study are a useful scientific contribution for decision-makers, which could help project this pre-treatment on a larger scale. Moving from laboratory scale to a larger one poses a challenge not only for science and engineering of FVBW drying processes but also a commitment for decision-makers, waste management stakeholders, and society as a whole. The average $M$ value reached after five days of treatment signifies easier handling, storage, and transportation of FVBW, which is desirable when destined for processes like biomass densification for energy conversion. An $M$ value of up to 10% in FVBW makes them more resistant to degradation by bacteria. Furthermore, since they acquire added value after drying, they become useful feedstock that could enter processes such as hydrothermal carbonization, torrefaction, and/or gasification. The FVWB resulting from drying could be integrated into closed-loop cycles if utilized in biorefineries, aiding in transitioning towards a circular economy rather than accumulating them in landfills, preventing the harmful effects that landfills pose to environment and society. Creating a beneficial endeavor centered on leveraging FVWB has the potential to yield societal and economic advantages within their generation, thereby contributing to cultivating a sustainable urban setting.

### Funding
This research received a grant from The National Council of Humanities, Sciences and Technologies (CONAYCYT) to Fernando Damián Barajas Godoy for his doctoral studies. The funders had no role in study design, data collection and analysis, decision to publish, or preparation of the manuscript.

### Grant Disclosures
The following grant information was disclosed by the authors:
The National Council of Humanities, Sciences and Technologies (CONAYCYT).

### Competing Interests
The authors declare there are no competing interests.

## Author Contributions

- Fernando Damián Barajas Godoy conceived and designed the experiments, performed the experiments, analyzed the data, prepared figures and/or tables, authored or reviewed drafts of the article, and approved the final draft.
- Marco A. Martínez-Cinco conceived and designed the experiments, analyzed the data, authored or reviewed drafts of the article, and approved the final draft.
- José G. Rutiaga-Quiñones conceived and designed the experiments, analyzed the data, authored or reviewed drafts of the article, and approved the final draft.
- Otoniel Buenrostro-Delgado analyzed the data, authored or reviewed drafts of the article, and approved the final draft.
- Jose Mendoza analyzed the data, authored or reviewed drafts of the article, and approved the final draft.

## Data Availability

The data is available at Zenodo: Barajas Godoy, F. D. (2024). Drying rate of biowastes [Data set]. Zenodo. https://doi.org/10.5281/zenodo.10636736.

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
