# Peer review of "The significance of biowaste drying analysis as a key pre-treatment for transforming it into a sustainable biomass feedstock"

_PeerJ, doi:10.7717/peerj.18248_

## Round 0.1 · original submission · Major Revisions

We have received positive feedback, however, some revisions are required.

Reviewer 1 ·

Basic reporting

The manuscript discusses the importance of bio-waste valorization and impact of drying parameters on valorization tendency. While the study addresses a valuable topic, there are several critical areas that need substantial improvement before its acceptance. Some suggestions for improvement are below

• Refinement of manuscript’s language and sentence structure is recommended to enhance readability, understanding, and quality.
• As abstract is standalone entity, so clear mention of study objectives and novelty is recommended for better context.
• Introduction should contain the information regarding “drying kinetics” and its importance in bio-waste valorization scheme.
• Novelty and objective of study has to be clearly mentioned in introduction for better context.

• Manuscript lacks the inclusion of recent references (specially from last 3 years), it is hereby recommended to add new and relevant references in subsequent sections.

Experimental design

No comments

Validity of the findings

• The essence of discussion is comparison of results (both qualitatively and quantitatively) with published literature, and currently this manuscript lacks sufficient references to analyze results. Revision of discussion section is recommended to contextualize findings and to highlight the significance of present study. Please include relevant literature and discuss how your results align with or differ from previous work.
• Conclusion should be presented as single paragraph, stating major findings and future outlook. Revise the conclusion for comprehensive representation of study.

Additional comments

• Revision of manuscript to avoid typing, generic, and grammatical mistakes is recommended to ensure the quality of presented data. Special attention should be given to line 1 (disposing off), line 73, line 93, line 326, and line 330.
• Sentence “by direct exposure to open sun of six batches” is hard to understand, rephrasing of said sentence is recommended.
• Extensive use of “the” is discouraged as its repetitive use makes manuscript less engaging. Limit its use to absolute necessary points is recommended.
• Sentence restructuring at line 70, line 80, line 81-83, 88-89, 95-96, 133-134, 328, 342, is recommended as presently it’s unable to convey the intended information.
• Use of proper punctuations is highly recommended to enhance coherence and readability. Special attention must be given to line 91.
• Italicize scientific names for correct representation.

Reviewer 2 ·

Basic reporting

The under-reviewed manuscript is focused on an important topic. The data presentation is nice. Overall, it is well-written. However, the language of the manuscript needs a careful recheck/revision to make things more concise/clear. Sometimes, sentences are too long and are unnecessarily broken into sub-sections.

Experimental design

The experimental design is satisfactory.

Validity of the findings

The findings seem rational and justified

Additional comments

1. Make sure the consistency of units throughout the manuscript
2. The consistency of the biowaste term should be maintained. “biowaste” or “bio-waste” follows one
3. Make sure that all abbreviations are stated in full form for the first-time use
4. Please use Insert Symbol options to insert various symbols, for example, the Degree sign for the temperature
5. Please make sure that all superscripts/subscripts are properly formatted (5th, 16th in the abstract)
6. Abstract: “Tow-term model” or “two-term model”?
7. The title is too long; it should be shortened, and it should be more elaborate and specific. For instance, “Contributing to the transition towards a circular economy” is not main study here, authors did not perform any experiments on that; it is a general approach. This part could be deleted from the abstract
8. “Bio-energy” and “bio-products” are recently written as “bioenergy” and “bioproducts”
9. Line 118-119. A full stop is added before and after the Citation. Please add a full stop after the reference only to ensure that the reference is related to the sentence before the reference. Please check it at other places, too
10. Did you observe the environmental impact of the drying process? Because it was in the open environment, any bad smell? Because smell means gases, gases could be a problem for the urban environment, imagine on a large scale.
11. The sample from various income categories could affect the process. The authors considered that and selected samples from a lower-middle stratum only. Why was only one economic class selected? Please elaborate on this in the manuscript
12. Drying was performed during June and July only. Weather conditions will affect the drying procedure. How about other months? If authors plan to perform drying in June-July only, how and where the huge amount of the biowaste will be stored? Because, if left untreated/unattended, its composition will be adversely changed due to natural degradation, while most of the organic component will be lost as gases
13. Discussion should be improved; seems superficial at some points
14. Conclusion is little longer than standard, should be more concise. Do not repeat results here. Give some concluding remarks based on the findings. Add some prospects and recommendations too.

---

## Round 0.2 · accepted · Accept

The revised version is accepted for publication!

Reviewer 1 ·

Basic reporting

Authors address all the comments.
1. Just change the starting line of abstract.
2. At line 250, there is typing mistake, correct "tow" to "two".

Experimental design

Rlevant and sufficient information.

Validity of the findings

All good

Additional comments

None

Reviewer 2 ·

Basic reporting

OK

Experimental design

OK

Validity of the findings

OK